# Progress toward SHAPE Constrained Computational Prediction of Tertiary Interactions in RNA Structure

**DOI:** 10.3390/ncrna7040071

**Published:** 2021-11-05

**Authors:** Grégoire De Bisschop, Delphine Allouche, Elisa Frezza, Benoît Masquida, Yann Ponty, Sebastian Will, Bruno Sargueil

**Affiliations:** 1Université de Paris, CNRS, UMR 8038/CiTCoM, F-75006 Paris, France; gregoire.de.bisschop@ircm.qc.ca (G.D.B.); delphine.allouche@inserm.fr (D.A.); elisa.frezza@u-paris.fr (E.F.); 2Institut de Recherches Cliniques de Montréal (IRCM), Montréal, QC H2W 1R7, Canada; 3Institut Necker-Enfants Malades (INEM), Inserm U1151, 156 rue de Vaugirard, CEDEX 15, 75015 Paris, France; 4Université de Strasbourg, CNRS UMR7156 GMGM, 67084 Strasbourg, France; b.masquida@unistra.fr; 5Ecole Polytechnique, CNRS UMR 7161, LIX, 91120 Palaiseau, France; yann.ponty@lix.polytechnique.fr (Y.P.); sebastian.will@lix.polytechnique.fr (S.W.)

**Keywords:** ribozyme, shape probing, RNA structure modeling

## Abstract

As more sequencing data accumulate and novel puzzling genetic regulations are discovered, the need for accurate automated modeling of RNA structure increases. RNA structure modeling from chemical probing experiments has made tremendous progress, however accurately predicting large RNA structures is still challenging for several reasons: RNA are inherently flexible and often adopt many energetically similar structures, which are not reliably distinguished by the available, incomplete thermodynamic model. Moreover, computationally, the problem is aggravated by the relevance of pseudoknots and non-canonical base pairs, which are hardly predicted efficiently. To identify nucleotides involved in pseudoknots and non-canonical interactions, we scrutinized the SHAPE reactivity of each nucleotide of the 188 nt long lariat-capping ribozyme under multiple conditions. Reactivities analyzed in the light of the X-ray structure were shown to report accurately the nucleotide status. Those that seemed paradoxical were rationalized by the nucleotide behavior along molecular dynamic simulations. We show that valuable information on intricate interactions can be deduced from probing with different reagents, and in the presence or absence of Mg^2+^. Furthermore, probing at increasing temperature was remarkably efficient at pointing to non-canonical interactions and pseudoknot pairings. The possibilities of following such strategies to inform structure modeling software are discussed.

## 1. Introduction

RNA molecules are ubiquitous within the cell but are also present outside of the cell as in plant phloem [1]. Some of them have even been shown to be glycosylated and exposed on the cell surface [2]. They fulfil very diverse functions including coding and non-coding roles, are involved in most steps of the genetic expression regulation, and can specifically interact with small molecules, proteins, or other nucleic acids (DNA or RNA). Their sequence is indisputably critical for the coding role or the specific recognition of other DNA and RNA sequences. Beyond the importance of their sequence, they also adopt specific three-dimensional structures that are responsible among others for the catalytic activity of ribozymes (RNA enzymes), the strong and specific binding to metabolites, proteins and ribosome, the assembly of phase-separated membrane-less organelles [3,4], the availability of sequence signals (e.g., [5]) and the modulation of ribosomes progression. Structure can be considered as another layer of encoded information, not only in compactly folded RNA, such as rRNA, tRNA, ribozymes, riboswitches but also within mRNA [6]. This is particularly obvious within viruses’ genomes in which information is very streamlined [7,8]. Given the central role of RNA and its structure, it is of great interest to be able to model RNA structure as accurately as possible. Physical methods that were very successful for determining protein structures suffer from limitations for RNA, for instance NMR can only be applied to short RNA and flexible RNA molecules are most of the time hardly amenable to crystallization for X-ray diffraction. In addition, those methods are very time consuming, while new RNA sequences are uncovered every day. The RNA structure modeling process most of the time firstly consists of predicting the secondary structure before inferring the three-dimensional architecture. However, identifying the correct secondary structure is complicated since the native structure is the result of indirect folding trajectories forming secondary and tertiary interactions. This is well exemplified by the formation of the P4P6 domain from the *Tetrahymena thermophila* group I intron. The secondary structure of the folding intermediate presents a P5abc domain with additional Watson–Crick base pairs as compared to the native form. The formation of tertiary interactions between P5abc and the P4 domain leads to the release of the additional base pairs that were yet required to direct the formation of the tertiary structure [9].

The accuracy of computational secondary structure modeling has been significantly improved due to the integration of experimental information from chemical structure probing in the form of soft constraints [10,11]. Two main types of chemical probes are used, those such as DMS (dimethyl Sulfate) or CMCT (cyclohexyl-3-(2-morpholinoethyl) carbodiimide metho-*p*-toluene sulfonate), that probe the availability of H-bond donor or acceptor sites located on the Watson–Crick face of the base [12], and SHAPE probes that report on the flexibility of the ribose [13,14,15,16]. Tremendous efforts and progress have been made over the last years to develop high throughput experimental [17,18,19,20,21] and computing pipelines [22,23].

However, the accurate prediction of the secondary structure remains a challenge for long RNA because of intrinsic RNA properties and technical problems. Firstly, most RNA are likely to adopt several conformations whether in vitro or in a cellular context, which may in some instances reflect experimental artefacts, but also reflect their potential to regulate their function through structural rearrangement. RNA structure probing experiments then yield averaged signals representing all the conformations and therefore do not properly inform the prediction software. Secondly, RNA structure includes a handful of non-canonical base pairings accounting for a significant part (≈25%) of the total number of base pairs [24] that significantly contribute to the folding stability, often by forming specific structural motifs. The relevance of such observation is striking in the case of the P4P6 domain of the *Didymium iridis* lariat-capping ribozyme (DiLCrz), the ribozyme used in this study. The DiLCrz crystal structure [25] shows a P4P5 secondary structure quite different from the originally predicted one [26]. In the original secondary structure, a four base pairs core is followed by two more taking place after a bulging U, whereas the crystal structure reveals that only one non-canonical base pair takes place after the bulge. The loss of the two WC base pairs is energetically compensated by the formation of a trans WC base pair, which fosters A residues from J5/4 to interact with the shallow groove of the P4 stem to weave A-minor interactions. In other words, the loss of secondary structure interactions is compensated by formation of tertiary ones. Such examples showcase that modeling of non-canonical and [canonical and non-canonical) tertiary interactions is an essential part of meaningful RNA structure modeling, even if such complex interactions add fundamental challenges in comparison to computational or computer-assisted modeling of canonical RNA secondary alone. Note that thermodynamics models and calculations without such interactions (i.e., restricted to canonical, secondary interactions) are long established and highly efficient for computer-assisted secondary structure prediction [27,28,29]. We can distinguish several lines of computational structure modeling including non-canonical and tertiary interactions. First, there are approaches that model RNA 3D structure, e.g., by full-atom fragment-based sampling methods [30,31] or coarse-grained simulation [32], which then allows reading-off secondary and tertiary interactions from the 3D representation. Typically, such methods are computationally limited, e.g., it is reported that “problems involving de novo building more than 80 nucleotides will be challenging for FARFAR2 conformational sampling” [30] whereas successful predictions are reported for up to 500 nt by RNAvista [33,34], which integrates fragment-based modeling with base pair annotation.

Other approaches like MC-Fold/MC-Sym [35], CycleFold [36], and RNAwolf [37] include non-canonical secondary interactions in their prediction of extended secondary structure. Such methods have the potential to predict larger structures, but (by principle) they do not predict tertiary interactions. The exact prediction of tertiary interactions has been well studied in the form of pseudoknot prediction. Pseudoknot motifs, which frequently occur in functional RNAs, are characterized by a short Watson–Crick helix, the pseudoknot, the strands of which are topologically separated: one of its two strands is intercalated between the strands of at least one other helix of the RNA. Because their prediction is computationally intractable in realistic energy models (NP-hard, inapproximable) [38,39], efficient algorithms were presented only for restricted classes of pseudoknots (e.g., [40,41]) and the most popular structure modeling methods ignore them completely. Nevertheless, pseudoknot prediction has been implemented in several faster algorithms [42,43] that first calculate pseudoknot free structures and then search, in a second round, for pairings between the still single-stranded bases, or which stochastically simulate the folding of the RNA chain in the course of virtual transcription [44]. Although quite successful, this strategy faces two pitfalls, first it generates several models that the user cannot discriminate, secondly, it relies on the accurate prediction of the secondary structure in the first round which is not certain for the reasons mentioned above. Although experimental probing data significantly improve structure prediction, it is not of much help for non-canonical interactions or pseudoknot pairings. Indeed, as they are involved in genuine Watson–Crick base pairs, pseudoknot positions show probing patterns similar to regular helices, while tertiary interactions are often mildly reactive as they are usually more dynamic than canonical pairings. Despite the above-mentioned difficulties, teams of experts gathering many competencies are able to generate impressively accurate tertiary structure models for compactly folded RNA [45,46,47].

Here we investigate several approaches in order to precisely identify nucleotides involved in tertiary structure elements, notably pseudoknots and kissing loops. To this end we considered two physicochemical properties of these structures: first, their formation is stabilized by the presence of magnesium and second, they have lower thermal stability than the core secondary structure. Our mid-term goal is to develop an experimental and computational workflow that allows any molecular biology laboratory to automatically model RNA secondary structure including kissing loops and pseudoknots and indicate the presence of other tertiary interaction. We take advantage of the well-established crystal structure of DiLCrz to serve as a benchmark for our study. This compactly folded RNA gathers in only 188 nucleotides many of the recurring structural motifs found in RNA including a pseudoknot, a kissing loop pairing, and tetraloop/receptor interaction.

## 2. Material and Methods

### 2.1. RNA Preparation

RNA was in vitro transcribed by run-off transcription using T7 polymerase in 40 mM Tris-HCl pH 8.0, 25 mM MgCl_2_, 5 mM DTT, 5 mM rNTPs, 1 mM spermidine and 20 U RNAsin (Promega, Madison, WI, USA). Template DNA was then digested at 37 °C for 20 min with RQ1 DNAse (Promega) and RNA was precipitated by addition of 2.5 M lithium chloride and centrifugation at 16,100× *g* for 30 min at 4 °C. RNA pellet was washed with 70% ethanol then resuspended in nuclease-free water. RNA was purified through G25 size exclusion chromatography and quantified spectrometrically. Its integrity as well as the absence of aberrant products were confirmed by gel electrophoresis.

### 2.2. Melting Curves

Six pmoles of RNA were diluted in 14 µL of H_2_O and denatured for 2 min at 80 °C. 4 µL of pre-warmed 5X folding buffer (HEPES pH 7.5 40 mM, KCl 500 mM, MgCl_2_ ranging from 0 to 5 mM final concentrations) were added and the solution was allowed to cool to room temperature in 5 min. After the addition of 2 µL of RiboGreen (ThermoFisher, Waltham, MA, USA, final concentration: 300 nM), the samples were incubated for 10 min at 37 °C. The melting curves were obtained with a PikoReal RT PCR system (ThermoFisher) by heating the sample from 37 to 95 °C at 0.04 °C·s^−1^ and measuring the fluorescence emitted by the ribogreen. Raw fluorescence curves were normalized and derived. Derivatives from three replicates were averaged and smoothed. Data analysis was performed using Graphpad Prism v5.02.

### 2.3. SHAPE Probing

*SHAPE chemical probing* was mostly performed as previously described in [48,49]. Briefly, 36 pmoles of DiLCrz RNA were diluted in 160 µL of water and denatured for 2 min at 80 °C. Then, 40 µL of pre-warmed folding buffer (HEPES pH 7.5 40 mM, KCl 500 mM, 5 mM MgCl_2_) were added and the solution was allowed to cool down to room temperature for 5 min. The sample was then incubated at 37 °C. After 10 min, a 20 µL aliquot (corresponding to 6 pmoles) was removed and added to 2 µL of 40 mM 1M7 and allowed to react for 2 min, while a second 20 µL aliquot was mixed with neat DMSO. The temperature was then gradually (about 10 min between each step) increased to 53 °C, 65 °C, 74 °C and 85 °C and the same probing procedure was applied. Probing with NMIA was performed identically, BzCN probing was performed at a final concentration of 40 mM and HEPES was adjusted to 80 mM. Probed RNA were precipitated at −20 °C with 10% ammonium acetate 5 M, 2.5 vol. ethanol and 20 µg of glycogen as a carrier. The pellets were washed with ethanol 70%, vacuum-dried for 10 min and resuspended in 10 µL of water. For the reverse transcription, probed RNA were denatured for 3 min at 95 °C after the addition of 1 µL DMSO. 6 pmoles of fluorescent primers (WellRed D2 or D4 fluorophore, Sigma-Aldrich, Saint Louis, MO, USA) (5′-D2 or D4 CTG-TGA-ACT-AAT-GCT-GTC-CTT-TAA-TG-AAG-TAT-TTG-AGG-TGT-AGA-GTG-TTT-GAG-TAG-TAG-TGA-TTG-TCT-TGG-GAT-ACC-GGA-3′) were then added and allowed to hybridize by incubating at 65°C for 5 min then at 35 °C for 10 min. Finally, after the addition of 0.5 mM of each deoxynucleotide (guanine substituted with inosine) plus the MMLV reverse transcriptase (RNAse H (-)) and its buffer (Promega), the elongation step was achieved by incubating the samples for 2 min at 35 °C, 30 min at 42 °C and 5 min at 55 °C. The sequencing reaction was performed with 3 pmoles of untreated RNA, D2-labeled primer and 0.5 mM of dideoxyadenosine triphosphate during the elongation.

Fluorescently labeled cDNA were then precipitated at −20 °C with 10% sodium acetate pH 5.4, 20 µg of glycogen as a carrier and 2.5 vol. of ethanol. Pellets were washed with ethanol 70%, vacuum-dried for 10 min, resuspended in 40 µL of sample loading solution and run on a CEQ 8000 capillary electrophoresis sequencer (Sciex, Concord, ON, Canada). The resulting traces were analyzed using QuSHAPE [50].

Reactivities were the mean from three independent replicates. Within each triplicate, values more than 0.4 reactivity units apart from the other measurements were considered as outliers and discarded.

### 2.4. Comparison of Two Probing Profiles

Probing profiles were compared as described in [49]. In brief, reactivities averaged over three replicates in two conditions, R1 and R2, were used to calculate the absolute difference ΔR = |R_1_ − R_2_| and the relative change ΔR/(R_1_ + R_2_). The reactivity difference between two conditions is considered significant when both variables exceed a threshold of 0.2, and the *p*-value from a two-sided *t*-test is inferior to 0.05. Nucleotides with undetermined reactivity are excluded from the analysis. Such an approach excludes small reactivity changes between weakly reactive nucleotides as well as large yet unmeaning reactivity changes between highly reactive nucleotides.

### 2.5. Nucleotide Clustering

*Nucleotides* were clustered by the KMeans algorithm as vectors of reactivities obtained at 37, 53, 65, 74 and 85 °C. Clustering was done using scikit-learn and a custom python script available at https://github.com/gdebissc/DiLCrz, accessed on 3 November 2021. Data were log2-transformed and standardized (nucleotide-wise then temperature-wise). Undetermined reactivities were removed prior to clustering. Dimensionality reduction by principal component analysis reveals that nucleotides can be grouped in at least three or four clusters. The optimal number of clusters was further determined using the elbow method. Briefly, the within-cluster sum of squared errors, or distortion, was calculated for each cluster number between 1 and 10 (Appendix A). The result graphically indicates that increasing the cluster number above four does not significantly reduce the distortion.

### 2.6. All-Atom Molecular Dynamics Simulations

MD simulations were carried out using GROMACS 5 package [51,52,53,54] using the Amber ff99+ parmbsc0 force field [55,56]. The molecular systems were placed in a cubic box and solvated with TIP4P-EW water molecules [57]. The distance between the solute and the box was set to at least 14 Å. The solute was neutralized with potassium cations and then K^+^Cl^−^ ion pairs were added to reach the salt concentration of 0.15 M. We used the ion corrections of Joung et al. [58] as this force field has been shown to produce stable RNA structures [59]. The parameters for Mg^2+^ are taken from [60]. In our protocol, Mg^2+^ ions were first placed at known binding sites while the remaining ions randomly replaced solvent molecules to reach a concentration of 0.02 M [61]. Long-range electrostatic interactions were treated using the particle mesh Ewald method [62,63] with a real-space cut-off of 10 Å. The hydrogen bond lengths were restrained using P-LINCS [53,64], allowing a time step of 2 fs. The translational movement of the solute was removed every 1000 steps to avoid any kinetic energy build-up [65]. After energy minimization of the solvent and equilibration of the solvated system for 10 ns using a Berendsen thermostat (τ_T_ = 1 ps) and Berendsen pressure coupling (τ_P_ = 1 ps) [66], simulations were carried out in an NTP ensemble at a temperature of 300 K and a pressure of 1 bar using a Bussi velocity-rescaling thermostat [67] (τ_T_ = 1 ps) and a Parrinello–Rahman barostat (τ_P_ = 1 ps) [68]. During minimization and heating, all the heavy atoms of the solute were kept fixed using positional restraints. The length of the simulation was 1000 ns. To assess the stability of the trajectory, we verified that the root mean square deviation (RMSD) and the number of hydrogen bonds were stable along the trajectory.

### 2.7. Definition of the Parameters

To get insights into the relationship between the structure and the reactivity, we computed several quantities along the MD simulation. First, we computed the hydrogen bonds and the stacking for each frame using the software MC-Annotate [69,70,71]. The hydrogen bonds were classified using the Leontis/Westhof classification [24,72] and we detailed if the HBs are formed between the O2′ and an atom in the Watson–Crick (W), Hoogsten (H) or the B side (the position is between the W and H side) or between OP and an atom in these sides [70].

Second, we computed the multiplets (triplets, quadruplet, etc.) and the secondary structure for each frame using the software DSSR (dissecting the spatial structure of RNA) v.2 [73]. For each hydrogen bond and multiplet, we computed the percentage of the occurrence.

To take into account the ribose conformation, we described the sugar ring using pseudorotation parameters. Although there are four possible pseudorotation parameters for a five-membered ring [74], two, in particular, are useful to characterize the sugar conformation: the so-called phase (Pha) and amplitude (Amp). While the amplitude describes the degree of ring puckering, the phase describes which atoms are most out of the mean ring plane. We calculated these parameters using the expressions given below [75]:$$Amp=\sqrt{\left(a^{2}+b^{2}\right)}\;;\;Pha=\left(\frac{a}{Amp}\right)$$
where $a=0.4\sum_{i=1}^{5}v_{i}cos\left[0.8\pi \left(i-1\right)\right]\;$ and $b=-0.4\sum_{i=1}^{5}v_{i}sin\left[0.8\pi \left(i-1\right)\right]$, with *v_i_* the ring dihedral angle *i*. This approach has the advantage of processing the ring dihedrals *v*_1_ (C1′-C2′-C3′-C4′) to *v*_5_ (O4′-C1′-C2′-C3′) in an equivalent manner [76]. Conventionally, sugar ring puckers are divided into 10 families described by the atom which is most displaced from the mean ring plane (C1′, C2′, C3′, C4′ or O4′) and the direction of such displacement (*endo* for displacements on the side of the C5′ atom and *exo* for displacements on the other side). Using the Curves + program [76] for each simulation trajectory, for each nucleotide we computed the percentage of appearance for each family. In order to understand the interplay between the sugar conformation and the chemical reactivity, we grouped the sugar puckers into two large families. The sugar puckers C1′-*exo*, C2′-*endo*, C3′-*exo*, C4′-*endo* belong to the B-like family, while C1′-*endo*, C2′-*exo*, C3′-*endo*, C4′-*exo* belong to the A-like family.

Finally, as in our previous work [14], we computed a set of distances, valence angles and dihedral angles to capture the flexibility of DiLCrz RNA structure. For each quantity, we computed the average value and its standard deviation.

## 3. Results

### 3.1. Probing DiLCrz Structure in Presence or Absence of Mg^2+^ Ions

DiLCrz RNA structure was first probed with different SHAPE reagents, namely NMIA, 1M7 and BzCN in native-like conditions including 5 mM of magnesium ions (Figure 1A). The reactivity maps obtained are discussed in the light of the X-ray structure previously reported [25]. We use the following nomenclature: Px are helices numbered as in Figure 1, Lx are the corresponding terminal loops, Jx/y are the nucleotides joining helix x to helix y following a 5′ to 3′ orientation, and 3WJ is the P2/P2.1/P10 junction. The exact nature of all the interactions evidenced in the 3D structure is summarized in the Appendix A. Overall, the three reactivity patterns fit remarkably well with the three-dimensional structure. The crystal structure was accompanied by SAXS data that already showed the agreement of the conformation in solution with the crystal structure. All these remarks indicate that the dominant conformation of DiLCrz in solution corresponds to the reported three-dimensional structure. Indeed, most nucleotides predicted to be involved in cis Watson–Crick base pairing are not reactive. Exceptions to this trend correspond to the region U_47_-G_49_/A_143_, and to few positions at the edge of helices (G_12_ (P2) or U_128_-A_132_ (P8)), which appear to react with NMIA, 1M7 or BzCN although they are located within a supposedly stable stem (Figure 1A). Very reactive nucleotides are located in loops, while moderately reactive positions are mostly involved in non-canonical or long-range interactions. However, the reciprocal is not true, for instance, not all single stranded nucleotides are reactive (see for instance G_74_, G_87_ or A_101_), and the reactivity of many nucleotides involved in tertiary contacts is indistinguishable from the reactivity of the nucleotides involved in the secondary structure. Notably, most nucleotides involved in pseudoknots are not reactive, which is deceptive for modeling algorithms that cannot predict such type of interaction at the first intention. As previously described for NMIA and 1M6 with various RNA [77,78], the reactivity profiles are very similar for the three probes, although some nucleotides appear to have different susceptibility to the different reagents (see Figure 1A and Appendix A). Yet, as discussed below, no systematic rule informative for modeling algorithms could be deduced from comparing the different profiles.

Magnesium ions are known to be a key element for RNA tertiary structure stability and compaction [79,80,81,82]. In order to capture the magnesium-dependent folding of DiLCrz the RNA was probed with the three SHAPE reagents in the absence of MgCl_2_. Here again, although not identical, the reactivity patterns obtained are very similar. Many of the nucleotides showing a reactivity change upon Mg^2+^ addition are identical for the three different probes (Figure 1B and Appendix A). Most importantly, the vast majority of the less reactive nucleotides (27 out of 32, i.e., 84% for 1M7) in the presence of Mg^2+^ are either single-stranded, involved in the pseudoknot, the kissing loop or a non-canonical interaction according to the crystal structure (see Appendix A). Among the exceptions are nucleotides involved in the 2 bp P6 which would be predicted to be very unstable on its own and is stabilized by the tertiary structure. A_69_ or G_75_ involved in helix closing base pairs also appear more sensitive to the probe in the absence of magnesium ions as well as G_134_ which is also involved in a tertiary interaction. Interestingly enough, few positions appear more reactive in the presence of Mg^2+^, among which nucleotides in L5 and L8 have been observed to be the most mobile in the crystal structure. Forty-four nucleotides are only involved in tertiary contacts, i.e., participating in pseudoknot-type interactions or in non-canonical base pairs but not at the same time in a WC base pair (triple interaction). Amongst these, 19 are significantly less reactive in presence of Mg^2+^ towards one probe or another (15 with 1M7). In contrast the reactivity of only 7 (for the three probes) out of the 102 nucleotides only involved in regular helices are sensitive to the presence of Mg^2+^. However, if probing in the absence of magnesium fairly clearly highlights nucleotides involved in the three-dimensional folding, there is one notable exception. Indeed we noted that the reactivity of the 5′ strand of the P7 pseudoknot (G_111_-G_115_) remains essentially unaffected by the absence of magnesium while P7-3′ becomes very reactive. This suggests that the pseudoknot is destabilized in absence of Mg^2+^ and that under such conditions P7-5′ sequence adopts an alternative folding. Interestingly, unconstrained predictions using RNAfold [83] or RNAstructure [84] consistently yield models in which A_112_-G_115_ are base paired with C_70_-U_74_ (Appendix A).

Thus, albeit very informative, the comparison of SHAPE reactivity profiles in the presence and absence of magnesium proves, in this case, to be insufficient to pinpoint the nucleotides involved in kissing-loops and pseudoknots.

### 3.2. Structure Characterisation by Thermal Denaturation

In order to evaluate the thermal stability of the various three-dimensional structural elements and highlight the involved nucleotides, we followed DiLCrz thermal unfolding. We first set up a melting experiment using the RiboGreen fluorescent dye as described [85]. Ribogreen fluoresces upon specific binding to single strand RNA. Upon temperature increase the different structural elements unfold, more single strand RNA becomes available for ribogreen binding and consequently increases the fluorescence signal. However, in the meantime, RiboGreen binding to RNA is destabilized as the temperature increases. The fluorescence curve in function of temperature therefore results from a non-specific constant fluorescence decay inflected or even inverted by the local increase of RNA single strand availability. As for UV thermal melting results, the curve inflections are determined by plotting the derivative of the relative fluorescence units (RFU) over temperature (-dRFU/dt). In such a plot, maxima represent the melting temperature transitions. Using this technology, we followed DiLCrz thermal denaturation at increasing MgCl_2_ concentrations ranging from 0 to 5 mM (see Figure 2A and Appendix A). In absence of Mg^2+^, fluorescence regularly decreases until 62 °C with only a slight inflection at 50 °C, followed by a significant increase peaking at 75 °C. The 50 °C inflection becomes more pronounced as [MgCl_2_] increases, shifts to 53 °C in presence of 2.5 mM and is finally individualized as a 57 °C peak at 5 mM MgCl_2_. In the meantime, the 75 °C peak gradually shifts to a higher temperature to reach 81 °C at 5 mM MgCl_2_ where a slight inflection at 70 °C can also be observed. As the 57 °C peak builds up with the presence of magnesium ions, we interpret it as the cooperative unfolding of the tertiary structure while the large peak at the higher temperature would represent the cumulative melting of the multiple secondary structure elements. To confirm this hypothesis, we repeated the experiment using three variants designed to disrupt the tertiary structure, A_90_G, G_111_C and A_168_G. Mutation A_90_G disrupts the triple interaction A_90_/G_72_-C_94_ that stabilizes P5-P4-P6 stacking, G_111_C destabilizes the P7 pseudoknot, and finally, A_168_ interacts with A_110_ and is in the heart of the catalytic center since the branching reaction occurs between C_167_ and U_169_ [86]. As can be observed in Figure 2B, in the presence of magnesium the 57 °C peak is almost or even completely abolished for the three mutants while the 80 °C peak is present for all three mutants and essentially similar to what is observed with the WT. In absence of magnesium, the melting profiles of the WT and the three variant constructs are almost identical, showing a single major peak at 75 °C (Figure 2C). This strongly suggests that the 75–80 °C represents the secondary structure, stable in absence of magnesium ions, while the 57 °C peak can be attributed to the cooperative melting of the tertiary structure which is formed only in the presence of divalent ions.

These results indicate that the thermal stability of DiLCrz secondary and tertiary structures are in two distinct temperature ranges under our experimental conditions. This led us to measure the 1M7 SHAPE reactivity profiles along the thermal denaturation. We thus established the reactivity profiles at five different temperatures from the folded structure (37 °C) to the essentially denatured sequence (85 °C) going through the local minima 53 °C, 65 °C and 74 °C (Appendix A). As expected for a progressive unfolding process, the average reactivity gradually increases while the reactivity value dispersion decreases with temperature. This is consistent with all nucleotides becoming equally very reactive. Note that some nucleotides still appear poorly reactive at 85 °C. This is due to the reactivity calculation/normalization process and means that they are less reactive than the average, but not that they are unreactive. In order to get a clearer view of the behavior of each nucleotide along the temperature curve, we clustered their reactivity profile using a k-means algorithm (see material and methods). Four clusters define the following categories. The first (grey in Figure 3A) gathers nucleotides that are very reactive at 37 °C, and remain very reactive (although their absolute reactivity value goes down because of the reactivity normalization process [50]). The second cluster regroups nucleotides which become highly susceptible to 1M7 modification over 37 °C (red in Figure 3A), while the third cluster is constituted by nucleotides which become increasingly reactive with temperature starting at 53 °C (green in Figure 3A). Finally, the last category clusters positions that become increasingly reactive only over 65 °C (blue in Figure 3A). Annotating the DiLCrz structure scheme makes visible that essentially the four categories also coincide with the different types of structural elements (Figure 3A,B). Indeed, most nucleotides of the same structural element fall in one of the four categories which can roughly be defined as single strand nucleotides (grey), nucleotides involved in tertiary contacts (red), secondary structure (green), and finally the highly stable helix P2.1 (in blue). Note that although highly unstable the U_128_-A_132_ base pair is attributed to the blue cluster, this is an artefact and limitation of our clustering methods. As general trends, tertiary structure opens up cooperatively between 53 and 65 °C while secondary structure starts melting at 65 °C and some elements appear to be stable over 80 °C. Notable exceptions are nucleotides in P10 (U_42_, G_171_, G_172_), the 5′ part of P3 (G_61_) and the bottom of P15 (C_142_-A_143_) that cluster with the tertiary contact nucleotides (red in Figure 3A). Indeed, those nucleotides appear to be involved in peculiar or dynamic pairings (see below). As nucleotides clustered on the basis of a clear reactivity increase between 37 and 53 °C appear to be mostly involved in tertiary structure, we carried out a statistical analysis to identify the positions with a significantly increased reactivity over this temperature transition (see Figure 3C and material and methods). Interestingly this approach is more conservative than the clustering, highlighting most nucleotides involved in the tertiary structure (non-canonical base-pairs and long distance pairing involved in pseudoknots). Notably, nucleotides involved in both sides of the P7 pseudoknot and the P5-P2.1 kissing loop are emphasized by this strategy. In addition to information about the tertiary structure, SHAPE reveals that the different helices open up at different temperatures, but not always in the order predicted by thermodynamic calculations on isolated helices. For example, P15 (ΔG° = −8.1 kcal), which is predicted to be the second most stable secondary structure element after P2.1 (ΔG° = −10.7 kcal), is essentially accessible at 65 °C while nucleotides within P2.1 are not reactive before 85 °C. Even more surprising, P9 (ΔG° = −2.8 kcal) or even the two base pair helix P6, which is predicted to be hardly stable when isolated, are not highly reactive below 85 °C. This is a mere example that thermodynamic calculation on isolated duplexes poorly describes helices that are part of a larger structure.

### 3.3. Molecular Dynamics Analysis

The vast majority of the high SHAPE reactivities observed in the study described above reveal single-stranded positions, while nucleotides involved in the secondary structure are essentially unreactive. However, we were puzzled by the reactivity of some nucleotides involved in stable Watson–Crick pairings which also appear to be differentially susceptible to the different SHAPE probes, or to be unexpectedly reactive at 53 °C or in the absence of divalent cations. In order to get insight into these unexpected SHAPE reactivities and better interpret probing results, we resort to molecular dynamics (MD). We combined the different quantities computed along the MD simulations to shed light on some particular regions of DiLCrz structure. First, we focused on the nucleotides that for each pair of probes show a significant difference in chemical reactivity. The nucleotides U_83_ (the 3′ residue from L5) and A_161_ (the 3′ residue from L9) are both involved in non-canonical hydrogen bonds and at some point of the MD simulation they can form a triplet (26% of the time) and a quadruplet (97% of the time), respectively. Figure 4A–C shows some distinct conformations for the pairing of the nucleotides U_83_ and A_161_. The ribose of U_83_ not only adopts several conformations, in particular the C2′-endo and C3′-endo, but also some pseudo-rotation intermediates, which is a sign of medium to high reactivity [14]. Moreover, the global flexibility of the nucleotide can explain the high reactivity for the three probes. On the contrary, in the case of A_161_, the ribose puckering changes from the stable C3′-endo conformation to an intermediate one (C2′-exo), and is less flexible than the previous ones along the simulation.

Second, we analyzed the nucleotide G_49_ (P15) that has a low reactivity to NMIA and 1M7 and a high reactivity toward BzCN. This result suggests that this nucleotide assumes a transient state before reaching a final and stable conformation. This hypothesis is supported by the observation that along the MD simulation, on the one hand, the puckering of G_49_ can briefly adopt a C2′-endo conformation and on the other hand, the local flexibility suggests the presence of two possible local conformations. A similar trend has been observed for the nucleotide G_182_ that has also a low reactivity with NMIA and 1M7 and a medium reactivity with BzCN.

Third, we focused on P10 and the nucleotides upstream and downstream from this helix (see Figure 4D). The change in reactivity for the region A_41_-G_44_ may reflect the ability of these nucleotides to form triplets along the MD simulation. Although most of this region is not very dynamic, while A_41_ can form several non-canonical hydrogen bonds and at the same time the ribose adopts a C2′-endo conformation, the ribose of U_42_ often adopts intermediate conformations and the nucleobase stacks with A_146_(downward stacking between U_42_ and A_146_ which also forms an outward stacking with A_175_).

Finally, we analyzed the P15 region, G_46_-A_53_/U_137_-U_144_ (see Figure 4E,F). Both strands of the helix are relatively flexible, however, in the region G_46_-A_53_ the ribose moieties are on average more flexible than in a standard helix and often adopt the C2′-endo conformation. On the contrary, within the U_137_-U_144_ strand, only the ribose of U_144_ is in the C2′-endo conformation. Moreover, on the 5′ strand nucleotides G_46_ and G_49_ stack with non-adjacent nucleotides C170 and A142, respectively. G_46_ also forms a non-canonical hydrogen bond with C_118_ (O2′B sides) along the MD simulation. These observations could explain why its reactivity increases up to 53 °C, although it is seen as a secondary structure pairing.

### 3.4. Using Probing Data for Secondary Structure Modeling

The probing results performed with the different SHAPE probes at 37 °C in the presence, or absence of Mg^2+^ ions were used as constraints for secondary structure prediction software. Most of the secondary structure was correctly predicted by RNAstructure and RNAfold, which both do not predict pseudoknots. However, in most of the proposed models the P7 part of the pseudoknot nucleotides are incorrectly paired (see Appendix A). In many of the models, A_112_-G_115_ are paired to C_70_-U_73_ compromising not only P7 but also the prediction of P4 and P6. Such misprediction can be expected as the P7 pseudoknot nucleotides are unreactive to the SHAPE reagents in native conditions, and only the 3′ part of the pseudoknot becomes reactive in absence of Mg^2+^. In order to better inform the prediction software about the tertiary structure and pseudoknots, the reactivity profile obtained at 53 °C was integrated in two different ways. In a first approach, 1M7 reactivity data collected at 53 °C were directly used as constraints. In a second approach, the nucleotides with significantly augmented reactivity when raising the temperature from 37 to 53 °C, were assigned a reactivity value of 10 while the reactivity of other nucleotides was kept to their value obtained at 37 °C with 1M7, NMIA or BzCN. This strategy prevents the formation of secondary structure pairings with the nucleotides involved in pseudoknots but does not improve the overall prediction. For instance, in some of the resulting models, P3 is destabilized to the benefit of spurious pairings.

We then evaluated the performance of our recently developed workflow IPANEMAP, which extends RNAfold-based probing-informed modeling by sampling and clustering steps that allow us to take multiple sets of probing data into account [87,88]. When using 1M7 reactivities as constraints IPANEMAP yields two models almost equiprobable. The first is the model described above with scrambled P7 and P4 for the prediction obtained with RNAstructure or RNAfold. In the second model, most pairings are correctly predicted including P4 and three out of five of the nucleotides involved in the P7 pseudoknot (G_111_-C_113_) are left unpaired in the loop of an alternative P6 hairpin (Appendix A). Running IPANEMAP with multiple sets of constraints representing any combination of the data obtained with other reagents, and/or in any of the other conditions described does not improve the models. Interestingly, when informed with the reactivity data obtained using NMIA in the presence or absence of Mg^2+^ IPANEMAP yields one model which captures most of the secondary structure, but the two base-pair P6 (Appendix A).

In another approach, we sought to predict the P7 pseudoknot pairings from the experimental results. To this end, we generated artificial constraints files aiming at indicating nucleotides potentially involved in pseudoknots to the prediction software. Reactivity of positions for which the SHAPE values significantly vary between two conditions was set to −0.2 while the value for other nucleotides was set to 5. When using such artificial constraints obtained from the reactivity with or without magnesium no base pair was predicted independently of the prediction software utilized. In contrast, when following this strategy with the probing results obtained with 1M7 at 37 and 53 °C, the only pairings predicted form P7. We conclude that the temperature differential SHAPE strategy allows for the prediction of DiLCrz pseudoknot.

## 4. Discussion

Evaluation of the reactivity profiles regarding the X-ray crystal structure confirms that SHAPE probing very accurately reflects RNA folding. Nucleotides highly reactive are single-stranded, while positions not susceptible to modification are involved in canonical pairings. Finally, nucleotides showing a mild reactivity are most of the time engaged in dynamic tertiary pairings. Molecular dynamics simulation further shows that paradoxical reactivity observed for nucleotides within stable helices most probably reflects the fast equilibrium of these nucleotides between two conformations. We then followed two strategies to pinpoint nucleotides involved in tertiary structure. The differential Mg^2+^ SHAPE analysis selectively reveals many of the nucleotides involved in non-canonical interactions. As expected, nucleotides involved in pseudoknots are also more reactive in absence of Mg^2+^, although they are WC cis base pairs. However, if the 3′ strand of P7 is well highlighted in the absence of Mg^2+^, the 5′ strand remains poorly reactive toward all probes. As suggested by some of the models obtained, it may reflect the involvement of these strands in alternative structure in absence of Mg^2+^ since this region mostly relies on tertiary interactions. The temperature-dependent SHAPE probing seems to be promising to spot residues involved in long-range pairings forming pseudoknots.

Several studies have reported the effect of magnesium and/or temperature on RNA chemical reactivity [89,90,91,92,93,94,95]. We observed an asymmetric change in P7 reactivity when magnesium was omitted. This observation is reminiscent of what has been observed with the env25 pistol pseudoknot. In this case, the 5′ strand reactivity is rather insensitive to the presence of magnesium while the 3′ strand reactivity increases in the absence of magnesium [95]. It also resembles temperature-dependent probing of tRNA^Asp^ where one strand of the D-stem becomes reactive at a temperature higher than the second one [92]. The authors mentioned that purines may maintain stacking interactions in the absence of base-pairing better than pyrimidines, which may be due to their bicyclic chemical structure. In line with this hypothesis, the purine-rich strand is the most stable in each of the three examples of asymmetric change in reactivity discussed here (DiLCrz P7, pistol pseudoknot and tRNA^Asp^ D-stem). Although P7 pseudoknot reactivity is asymmetrically affected by magnesium, it is symmetrically affected by the increase of temperature. In fact, probing upon temperature ramping and probing in presence and absence of magnesium reflect well distinct processes. While in the magnesium-dependency experiment, the RNA is folded either in the presence or absence of magnesium, in the temperature-dependent experiment it is first folded at 37 °C then progressively denatured. Thus, in the first case, we observe two independent conformations while in the second case the successive conformations are dependent on the initial conformation.

### 4.1. DiLCrz Hierarchical Thermal Unfolding

The four different temperature clusters suggest somehow an unfolding model of the DiLCrz. Below 37 °C, the reactivity of P8 and P5 loops (L8 and L5) indicate intrinsic molecular motions that are observed in the crystal structure. While there is no electronic density for residues from L8 in the X-ray DiLCrz crystal structure, L5 is observed since the kissing interaction with L2.1 somehow maintains the L5 structure, in spite of greater than average temperature factors (B factors), which express the uncertainty of the atomic coordinates. Highly reactive residues from J6/7 (C_109_-A_110_) have also higher temperature factors. The same observation applies to residues from J15/7 (G_145_, A_146_) which interact with the three-way junction (P2/P2.1/P10) and are also characterized by higher B factors. At the same time, L9 is also reactive, indicating a strong cooperativity between the tertiary interactions L9-P2, L2.1-L5, and the J15/7-3wj interaction. The 3wj acts as a receptor for A_146_. This interaction is instrumental to stabilize the trans WC pair G_46_-U_144_. Heating beyond 37°C provokes melting of this G-U pair and other residues from cluster 2 (red). Opening up the P15 G-U pair is accompanied by melting of P10 and P7. Simultaneously, the tertiary interactions L9-P2 and L2.1-L5 are lost. The A-minor motif from J5/4 is also lost, breaking J3/4, which cannot stabilize P6 anymore. Then, when 65 °C is reached, residues from cluster 3 (green) start to melt. It is worth noting that, at this stage, the secondary structure elements that are intimately linked to the tertiary structure have already been partially lost, like P10 and P4P6. The fourth cluster regroups residues that remain folded in spite of a high temperature (blue). Surprisingly, the most stable element is P2.1. Since P2.1 is on the 5′ side of the ribozyme, it is transcribed early during transcription and is presumably the first element of the ribozyme to be folded. Moreover, this element, together with P2 and P10, constitutes a three-way junction, which binds A_146_ in J15/7, a critical region for catalysis, thus potentially regulating the activity of the ribozyme [86]. The present data thus suggest that P2.1 may be a folding nucleation element toward the active structure of the ribozyme.

The present scheme is supported by previous studies on the tRNA^Asp^ thermal unfolding using NMIA [92]. The authors identified five melting transitions that can be clustered in three main unfolding events. The initial event is described by the first two transitions (51 and 53 °C, respectively), first concerning the melting of the tertiary interactions mediated between the D- and T-loops, secondly followed by D-stem strands separation (53 °C). The second event at 58 °C mainly consists in melting of the acceptor and anticodon regions, which reorganize by forming alternative pairings between the 3′ residues from the acceptor stem, the 5′ strand form the anticodon stem, and the 3′ end from the anticodon loop. The formation of this alternative secondary structure presumably compensates for the tertiary structure loss. The third event occurs when the alternative stems melt individually when the temperature rises. The T-stem loop remains incredibly stable along the whole process and may be compared to P2.1 from DiLCrz. Although we cannot exclude that alternative secondary structures could be formed during thermal unfolding of DiLCrz, the present SHAPE data do not present differential reactivities between cognate helical strands at temperatures <65 °C. The DiLCrz unfolding scheme thus appears as a more hierarchical process than the one from tRNA^Asp^.

### 4.2. Molecular Dynamics Shed Light on Unexpected Reactivity

As previously reported, the different SHAPE probes show very similar but not identical reactivity patterns [77,96]. The analysis of the structure dynamics of DiLCrz shows that when significant differences are observed between the probes, the reactivities are medium to high, and the nucleotides are most of the time involved in multiplets. For example, the nucleotide A_161_ is mostly involved in a quadruplet and there are significant differences between each pair of probes. Another example is given by the nucleotide C_171_ that is involved in either a triplet or a multiplet with four other nucleotides.

In the case of G_182_ where low reactivity towards both NMIA and 1M7 was observed, while a high/medium reactivity was observed with BzCN, the nucleotide adopts a transient conformation before reaching a final and stable state (see Appendix A). Indeed, along the MD simulation the puckering of the nucleotide briefly adopts a C2′-endo conformation, changing from C3′-endo and some intermediate states close to either the C2′-endo or the C3′-endo conformation. In addition, the structural local flexibility suggests the presence of two possible conformations.

Finally, our study confirms that the flexibility of the sugar is a key element for the chemical reactivity but that further investigations are still required to better characterize the SHAPE reagent reactivity and clarify what appears to be a complex scenario. For instance, the high to medium reactivity observed with both NMIA and BzCN within P15 were quite puzzling and could lead one to question a model displaying P15. Molecular dynamics suggests that albeit stable this helix is flexible, showing transient alternative conformation and unusual stacking.

### 4.3. Informing Prediction Software to Integrate Pseudoknots and Other Tertiary Motifs

Although pseudoknots are present in many RNA, predicting them is still a challenge in terms of modeling and computation. Thus, as exemplified in this study, failing to take them into account may lead to incorrect secondary structure models. While more general and exact methods seem precluded, predicting pseudoknots in two rounds has been applied successfully; most prominently in ShapeKnots [43]: a first round predicts several models for secondary structure, and a second round searches for complementarity between the remaining single stranded nucleotides; identified pseudoknots are rewarded with a thermodynamic bonus. Using the 1M7 reactivity pattern obtained in this study, ShapeKnots delivers ten possible models among which the second most stable is a model including the P7 pseudoknot and otherwise correct but for the absence of P6 and the kissing-loop between P5 and P2.1. Despite this unquestionable success, in absence of additional data, it is impossible for the experimenter to decide which of the 10 models is the most likely. Data such as those presented here (Mg^2+^ or temperature differential SHAPE) undoubtedly indicate which model to choose. As shown above, the data obtained with the temperature differential SHAPE allow to identify unambiguously the P7 pseudoknot, and when constrained with the 1M7 or the NMIA reactivity map IPANEMAP yields models in which respectively, part or all of the P7 sequences are left available. The combination of both steps would lead to a model identical to the one obtained with ShapeKnots. However, such a post modeling evaluation for ShapeKnots or semi-automated modeling for IPANEMAP is feasible in the modeling of a short RNA such as DiLCrz, but does not appear realistic in the context of longer RNA. In the near future we will extend the IPANEMAP workflow to infer tertiary interactions based on multi-condition probing data. The system will be informed of nucleotides with a high probability to form tertiary structure. The software will at first exclude them from secondary structure modeling, which allows it to identify pseudoknot helices and even recurrent tertiary structure motifs in a second modeling phase. For instance, comparing the profiles of the three probes on DiLCrz highlights A_161_, a nucleotide part of a GAAA tetraloop that interacts with a receptor. Such motifs are frequent within RNA structures, the combination of multiple SHAPE probing and the signature sequence (GNRA) could help to identify such structures ab initio. One of the challenges in modeling DiLcRz structure is the prediction of the L2.1/L5 kissing loop. Both temperature- and Mg^2+^- differential probing strategies detect most of the nucleotides involved in the kissing loop (4 and 5 out of 6 nt, respectively). However the informed second round of modeling does not predict 2 bp long interactions to avoid the prediction of spurious pairings. In contrast, predicting three base pairs long range or cross interactions should be possible—it would require the prior detection of all the nucleotides involved. Comparing SHAPE data performed in different conditions or with different probes reveals not only the secondary but also some elements of the tertiary contacts. In this context, it is crucial to further understand the mechanisms behind SHAPE reactivities and the interplay between SHAPE probes, their reactivities, and RNA structures using methods at different length scales and by comparing diverse experimental data on a RNA benchmark dataset. This will allow deriving more general rules to inform 2D and 3D modeling software. In particular, the integration of this new information in an innovative theoretical framework for molecular dynamics simulations will open the route to bias three dimensional RNA structures and to better predict their structural motifs.

## Figures and Tables

**Figure 1 ncrna-07-00071-f001:**
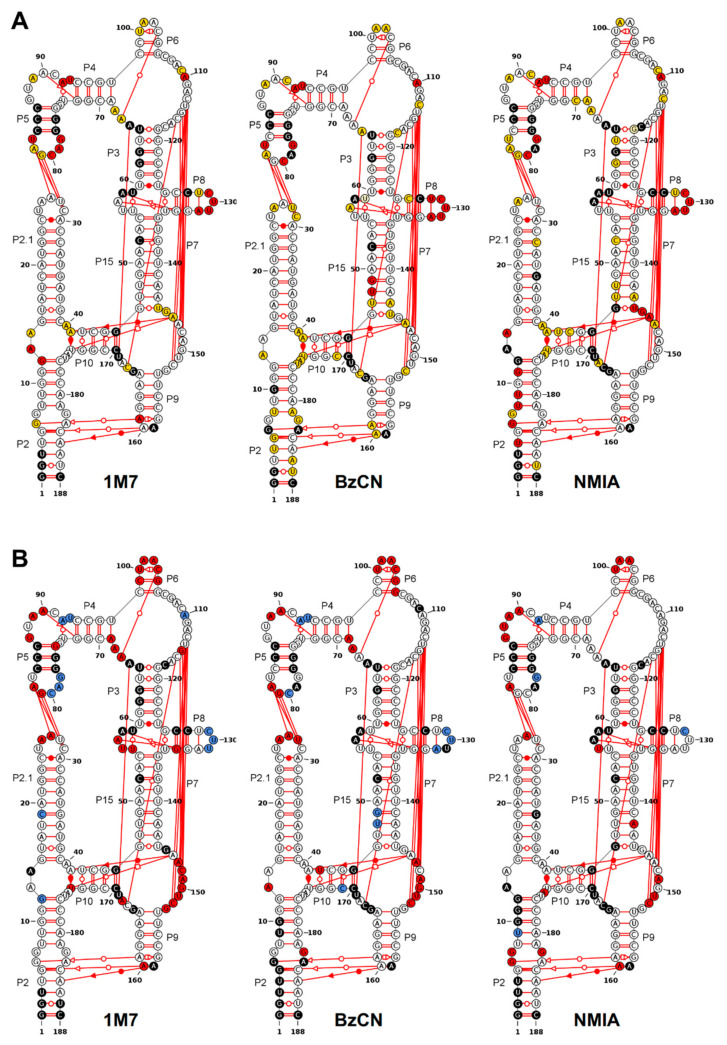
DiLCrz reactivity profiles with 1M7, BzCN and NMIA. 1M7, BzCN or NMIA were used to probe the structure in the presence or absence of magnesium. Probing was conducted at 37 °C in 40 mM HEPES pH 7.5, 500 mM KCl with or without 5 mM MgCl_2_. (**A**) Color coded reactivities obtained in the presence of magnesium pasted over DiLCrz secondary structure. White: low reactivity (0 to 0.4), yellow: moderate reactivity (0.4 to 0.7), red: high reactivity (over 0.7). (**B**) 1M7 reactivities obtained in the presence and absence of magnesium were compared as described in Material and Methods. Red: more reactive in the absence of magnesium. Blue: more reactive in the presence of magnesium. White: no significant difference. Nucleotides in black denote undetermined reactivity.

**Figure 2 ncrna-07-00071-f002:**
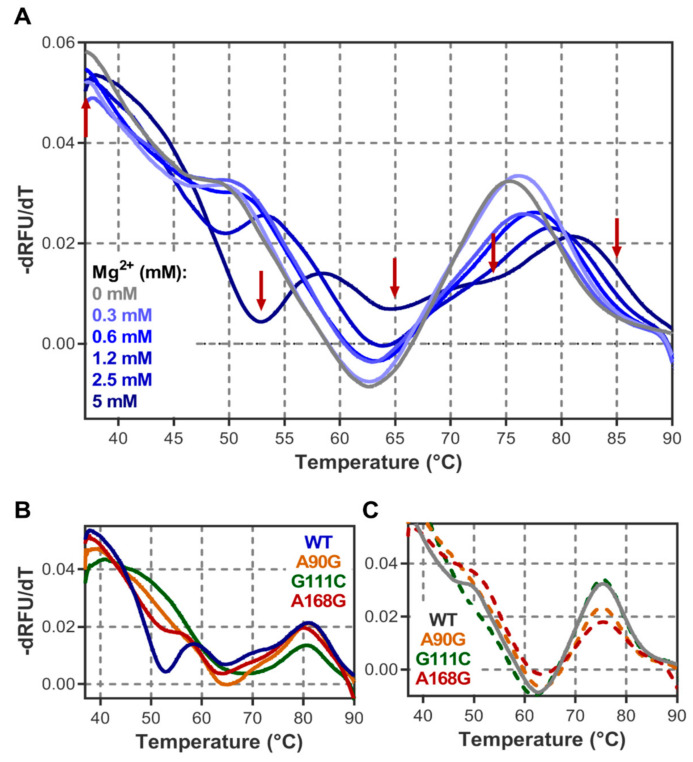
DiLCrz melting curves. DiLCrz equilibrated at 37 °C in 40 mM HEPES pH 7.5, 500 mM KCl and the indicated concentrations of magnesium was heated at 0.04 °C/s up to 95 °C, and melting curves were acquired using the RiboGreen dye fluorescence. (**A**) DiLCrz melting profiles in the function of Mg concentration. Gray: absence of magnesium; light blue to dark blue: 0.3, 0.6, 1.2, 2.5 and 5 mM MgCl_2_, respectively. Red arrows indicate the temperatures at which probing was conducted. (**B**,**C**) Effects of A_90_G, G_111_C and A_168_G mutations on the melting curves in the presence (**B**) or absence (**C**) of 5 mM MgCl_2_. Fluorescence derivatives were averaged from three independent experiments and smoothed.

**Figure 3 ncrna-07-00071-f003:**
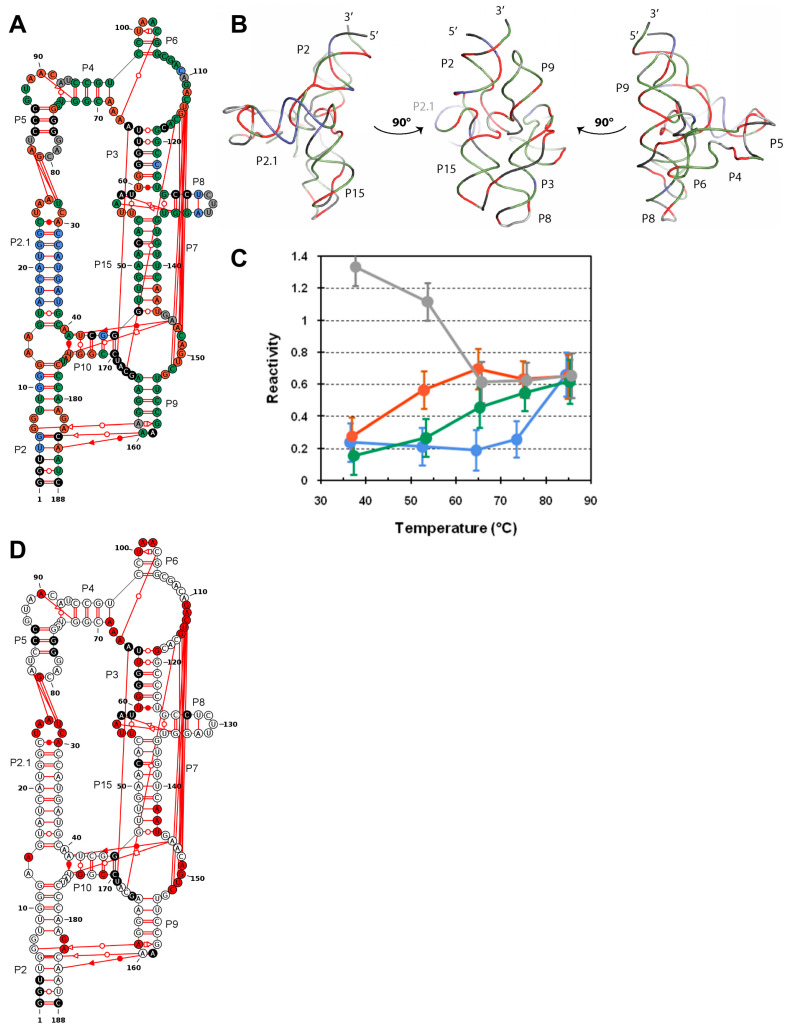
DiLCrz reactivity profiles at different temperatures. DiLCrz structure was probed with 1M7 at 37, 53, 65, 74 and 85 °C in 40 mM HEPES pH 7.5, 500 mM KCl and 5 mM MgCl_2_. (**A**) Clustering of the nucleotides according to their reactivity change. KMeans clustering was performed on 1M7 reactivities. DiLCrz secondary structure coloring refers to the nucleotides’ assigned clusters. (**C**) Previously reported DiLCrz X-ray structure [25] colored according to the clustering. (**B**) Evolution of the average reactivity within each of the four clusters. Error bars refer to the reactivity mean absolute deviation within each cluster. Points are horizontally jittered for clarity. (**D**) Comparison of 1M7 reactivities between 37 and 53 °C. Significant reactivity differences are highlighted in red. The comparison was performed as described in Material and Methods. Nucleotides in black denote undetermined reactivity.

**Figure 4 ncrna-07-00071-f004:**
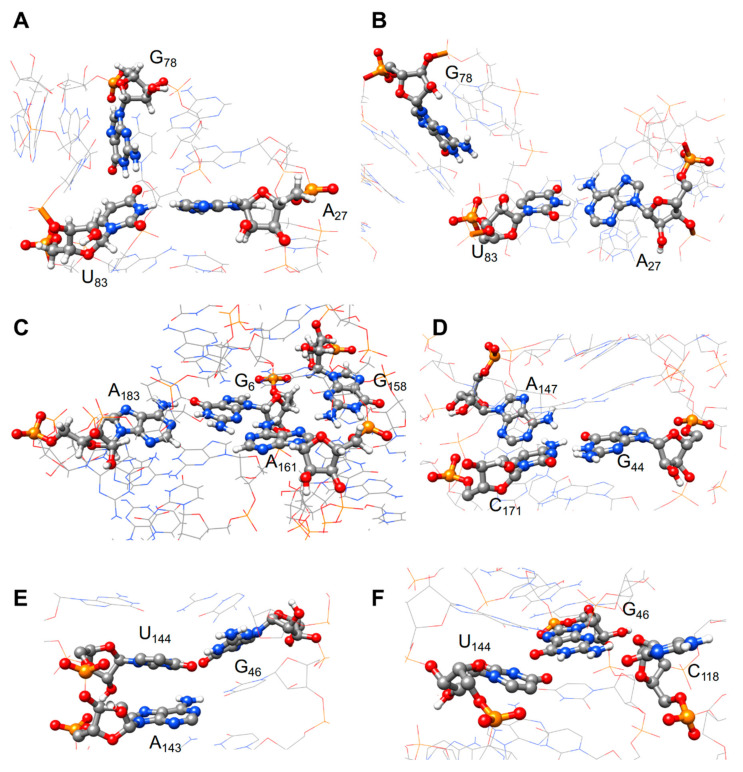
Different base pairing for some nucleotides of DiLCrz along MD simulation. Examples of different base pairing for the nucleotides U_83_, A_161_ and X obtained along the MD simulations. (**A**) Triplet (U_83_, A_27_ and G_78_). U_83_-A_27_: cWW. U_83_-G_78_: tWH. (**B**) Canonical base pairing between U_83_ and A_27_, the HB between U_83_, and G_78_ is lost. (**C**) Quadruplet (A_183_, G_6_, A_161_ and G_158_). G_158_-A_161_: tSH. G_6_-A_161_: tSW. G_6_-A_183_: cWW. (**D**) Triplet (C_171_, A_147_ and G_44_). G_44_-C_171_: cWW. C_171_-A_147_: cHW. (**E**) Triplet (U_144_, A_143_ and G_46_). G_46_-U_144_: cWW. U_144_-A_143_: H/O2′ (inward stacking). (**F**) Triplet (U_144_, C_118_ and G_46_). G_46_-U_144_: cWW. G_46_-C_118_: B/O2′.

## Data Availability

The molecular dynamic trajectory used in this work is available at the doi:10.5281/zenodo.5642809.

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
