# Peer review of "Progress toward SHAPE Constrained Computational Prediction of Tertiary Interactions in RNA Structure"

_ncrna, 2021, doi:10.3390/ncrna7040071_

Round 1

Reviewer 1 Report

The manuscript by De Bisschop et al. presents a thorough and complete analysis of the structural elements of a particular medium sized (~190 nt) folded RNA, the Didymium iridis lariat-capping ribozyme (DiLCrz). A high-resolution crystal structure of the ribozyme has been solved in the laboratory of one of the authors, and there is strong evidence from SAXS experiments that the crystal structure is an accurate representation of solution structure. With this data in hand, the authors use thermal melting and extensive SHAPE (Selective 2'-hydroxyl acylation analyzed by primer extension), and molecular dynamics experiments to derive a folding pathway for the ribozyme and to make correlations between results from SHAPE experiments and particular types of secondary and tertiary structures. The paper is a tour de force in its thorough analysis of the DiLCrz structure, and provides a much more complete understanding of the ribozyme than can be obtained from the crystal structure alone. For example, the clustering of nucleotides based on their temperature dependent SHAPE reactivity and the correlation with tertiary and secondary interactions is fantastic as a validation for the RNA structure.  

However, the paper does not quite arrive at the what the authors describe as their "mid-term goal" of creating a generalized workflow for the experimental and computational determination of RNA structure. This is especially true because the authors' experimental pipeline gives two potential structures with slight differences that can only be differentiated using the crystallographic data.  

My only suggestions for changes are as follows:

• Change the title of the manuscript. Something like "Further insight into the structure of the Didymium iridis lariat-capping ribozyme from variable temperature SHAPE probing" would be more appropriate. Alternatively, "Progress towards SHAPE constrained computational determination of RNA structure". Many other possibilities exist that would be considerably more informative than the current title. 

• Provide information in the abstract about the particular RNA that is being investigated.

• Include a section in the discussion that more clearly discusses how the lessons learned from this work are generally applicable to structure determination for other RNAs. Particular focus should be given to the kissing loop interaction. It seems that the authors hoped that SHAPE reactivities in the presence and absence of magnesium would be sufficient to identify nucleotides involved in this interaction, but that proved not to be the case in this RNA. (Line 331) Do the authors think that this situation is general? 

Author Response

We would like to thank the reviewers for their careful reading of our manuscript and their constructive criticisms. Please find below a point by point answer to their concerns, essentially the manuscript was amended as suggested by the reviewers. The manuscript was further carefully edited for typos, grammatical errors and sentences hard to understand.

We hope that the manuscript is now acceptable for publication in “non-coding RNA”.

 Reviewer 1 :

The manuscript by De Bisschop et al. presents a thorough and complete analysis of the structural elements of a particular medium sized (~190 nt) folded RNA, the Didymium iridis lariat-capping ribozyme (DiLCrz). A high-resolution crystal structure of the ribozyme has been solved in the laboratory of one of the authors, and there is strong evidence from SAXS experiments that the crystal structure is an accurate representation of solution structure. With this data in hand, the authors use thermal melting and extensive SHAPE (Selective 2'-hydroxyl acylation analyzed by primer extension), and molecular dynamics experiments to derive a folding pathway for the ribozyme and to make correlations between results from SHAPE experiments and particular types of secondary and tertiary structures. The paper is a tour de force in its thorough analysis of the DiLCrz structure, and provides a much more complete understanding of the ribozyme than can be obtained from the crystal structure alone. For example, the clustering of nucleotides based on their temperature dependent SHAPE reactivity and the correlation with tertiary and secondary interactions is fantastic as a validation for the RNA structure.  

However, the paper does not quite arrive at the what the authors describe as their "mid-term goal" of creating a generalized workflow for the experimental and computational determination of RNA structure. This is especially true because the authors' experimental pipeline gives two potential structures with slight differences that can only be differentiated using the crystallographic data.  

My only suggestions for changes are as follows:

  • Change the title of the manuscript. Something like "Further insight into the structure of the Didymium iridis lariat-capping ribozyme from variable temperature SHAPE probing" would be more appropriate. Alternatively, "Progress towards SHAPE constrained computational determination of RNA structure". Many other possibilities exist that would be considerably more informative than the current title. 

As suggested by the reviewer we have now changed the title to:

“Progresses toward SHAPE constrained computational prediction of tertiary interactions in RNA structure”

  • Provide information in the abstract about the particular RNA that is being investigated.

Instead of “benchmark RNA” it is now stated in the abstract “the 188 nt long lariat-capping ribozyme”

  • Include a section in the discussion that more clearly discusses how the lessons learned from this work are generally applicable to structure determination for other RNAs. Particular focus should be given to the kissing loop interaction. It seems that the authors hoped that SHAPE reactivities in the presence and absence of magnesium would be sufficient to identify nucleotides involved in this interaction, but that proved not to be the case in this RNA. (Line 331) Do the authors think that this situation is general? 

The section in the discussion that deals with the lessons learned from this study has been extended, with a specific discussion about the kissing loop interaction and why it was not identified. As now stated in the manuscript, kissing loop can be detected but 2 bp long distance interactions can not be predicted by such an approach because they will increase the risk of predicting spurious pairings.

Reviewer 2 Report

De Bisschop et al. presented an article about the relation of probing data to RNA 3D structure information. They selected DiLCrz, which contains a complex RNA architecture with junctions and pseudoknots, and performed many analyses on that data. They showed ways of incorporating probing data in varying conditions to improve the prediction of RNA tertiary interactions.

The article references an essential topic of RNA structure modeling. The authors incorporated several techniques: probing profiling, thermal denaturation, molecular dynamics, and computational analysis. I believe this manuscript is a valuable entry in the field, but I have some comments for the authors:

  • You mention MC-Fold, CycleFold, and RNAwolf for extended secondary structure prediction. There is also the RNAvista method, reported being superior to MC-Fold and RNAwolf in Rybarczyk et al. in New in silico approach to assessing RNA secondary structures with non-canonical base pairs. I think this deserves a note in the article as well.
  • The SHAPE probing subsection of Materials and Methods starts as if something is missing in the first sentence.
  • You used a lowercase "l" (ell) instead of a straight line "|" for the absolute reactivity difference formula
  • The Nucleotide Clustering subsection is not entirely clear. If I understand correctly, you were clustering tuples/vectors? Did you describe each nucleotide by several reactivity values taken at different temperatures? For example, a red cluster's exemplar might be [0.3, 0.6, 0.7, 0.6, 0.6] (values taken from Figure 3). Do I understand correctly? Can you improve this section to make it clear?
  • The ring dihedrals description misses the prime symbol. It is C1', not C1 (and so on for other atoms)
  • In the literature, the definitions of those dihedrals are different. v0 is C4'-O4'-C1'-C2', v1 is O4'-C1'-C2'-C3', etc. And people usually compute the phase and amplitude of the pseudorotation according to Altona and Sundaralingam with different formulas. Can you provide a citation or other rationale for the formulas you use? By the way, you probably mistyped them because, in the PDF, they display as "v_i cos cos (...)".
  • Lines 320-324: The lengthy sentence is unclear and hard to follow. Are 19 nucleotides "less reactive" or are "pointed by this process." Can you rephrase that sentence?
  • You write that the reactivity of P7 pseudoknot is unaffected by the absence of magnesium. You conclude that this suggests alternative folding. Maybe I misunderstood that part. I would assume that if reactivity is unchanged, the pseudoknot forms with or without magnesium ions. Can you clarify why alternative folding is your hypothesis?
  • Citation types are sometimes mixed. I found a citation in the form "(Meyer et al 2014)", while elsewhere, the citation format is a number in brackets.

Author Response

Dear Editor,

We would like to thank the reviewers for their careful reading of our manuscript and their constructive criticisms. Please find below a point by point answer to their concerns, essentially the manuscript was amended as suggested by the reviewers. The manuscript was further carefully edited for typos, grammatical errors and sentences hard to understand.

We hope that the manuscript is now acceptable for publication in “non-coding RNA”

De Bisschop et al. presented an article about the relation of probing data to RNA 3D structure information. They selected DiLCrz, which contains a complex RNA architecture with junctions and pseudoknots, and performed many analyses on that data. They showed ways of incorporating probing data in varying conditions to improve the prediction of RNA tertiary interactions.

The article references an essential topic of RNA structure modeling. The authors incorporated several techniques: probing profiling, thermal denaturation, molecular dynamics, and computational analysis. I believe this manuscript is a valuable entry in the field, but I have some comments for the authors:

  • You mention MC-Fold, CycleFold, and RNAwolf for extended secondary structure prediction. There is also the RNAvista method, reported being superior to MC-Fold and RNAwolf in Rybarczyk et al. in New in silico approach to assessing RNA secondary structures with non-canonical base pairs. I think this deserves a note in the article as well.

Please note that RNAvista was already mentioned in our submitted manuscript, although we agree that we did not provide much details. We modified the text as suggested by the reviewer, and mentioned the capacity of RNAvista to model larger RNAs.

  • The SHAPE probingsubsection of Materials and Methods starts as if something is missing in the first sentence.

We modified the text as suggested by the reviewer

  • You used a lowercase "l" (ell) instead of a straight line "|" for the absolute reactivity difference formula

We modified the formula as suggested

  • The Nucleotide Clustering subsection is not entirely clear. If I understand correctly, you were clustering tuples/vectors? Did you describe each nucleotide by several reactivity values taken at different temperatures? For example, a red cluster's exemplar might be [0.3, 0.6, 0.7, 0.6, 0.6] (values taken from Figure 3). Do I understand correctly? Can you improve this section to make it clear?

Yes the reviewer is right. We now clarified the first sentence of this subsection.

  • The ring dihedrals description misses the prime symbol. It is C1', not C1 (and so on for other atoms)

We thank the reviewer for highlighting the typos, they were corrected (see page 5).

  • In the literature, the definitions of those dihedrals are different. v0 is C4'-O4'-C1'-C2', v1 is O4'-C1'-C2'-C3', etc. And people usually compute the phase and amplitude of the pseudorotation according to Altona and Sundaralingam with different formulas. Can you provide a citation or other rationale for the formulas you use? By the way, you probably mistyped them because, in the PDF, they display as "v_i cos cos (...)".

We thank the reviewer for highlighting the typos. We modified the equation and added a reference for our definition.

  • Lines 320-324: The lengthy sentence is unclear and hard to follow. Are 19 nucleotides "less reactive" or are "pointed by this process." Can you rephrase that sentence?

Thanks for pointing out to this unclear sentence. It has now been rephrased to “Forty-four nucleotides are only involved in tertiary contacts, i.e participating in pseudoknots-type interactions or in non-canonical base pairs but not at the same time in a WC base pair (triple interaction). Amongst these, 19 are significantly less reactive in presence of Mg2+ towards one probe or another (15 for 1M7 ).In contrast the reactivity of only 7 (for the three probes) out of the 102 nucleotides only involved in regular helices are sensitive to the presence of Mg2+.”

We hope that the sentence better reads now

  • You write that the reactivity of P7 pseudoknot is unaffected by the absence of magnesium. You conclude that this suggests alternative folding. Maybe I misunderstood that part. I would assume that if reactivity is unchanged, the pseudoknot forms with or without magnesium ions. Can you clarify why alternative folding is your hypothesis?

We probably were not clear enough about P7 pairing: the 5’ strand of the paring does not become more reactive, but the 3’ strands become very reactive showing that the P7 pairing is destabilized in absence of Mg. This has been clarified in the text.

  • Citation types are sometimes mixed. I found a citation in the form "(Meyer et al 2014)", while elsewhere, the citation format is a number in brackets.

The manuscript has been carefully checked for typos, misspelling, problems with references, we hope that it is OK now.